# Investigation of Crack Repairing Technique to Delay Fracture Initiation of Steel Members Subjected to Low Cycle Fatigue

**Sampath Abeygunasekara** [1,*], **Jeeva Chandanee Pushpakumari Gamage** [1] **and Sabrina Fawzia** [2]

1 Department of Civil Engineering, Faculty of Engineering, University of Moratuwa, Katubedda 10400, Sri Lanka; kgamage@uom.ac.lk
2 School of Civil & Environmental Engineering, Queensland University of Technology, Brisbane 4000, Australia; sabrina.fawzia@qut.edu.au
* Correspondence: abeygunasekarasampath@gmail.com

**Abstract:** Stress concentrations have become a common phenomenon in steel elements when arresting a fracture by implementing the crack stop hole (CSH) technique. Embedding the CSH with Carbon Fibre-Reinforced Polymer (CFRP) enhances the fatigue life by delaying fractures while achieving stiffness recovery due to the superior mechanical characteristics of the CFRP material. Hence, the low cyclic fatigue (LCF) behaviour of 90 strengthened and non-strengthened CSH specimens was examined in this context. These specimens were subjected to a range of 0 to 10,000 fatigue load cycles at a frequency of 5 Hz. At the end of fatigue exposure, the average tensile strength was measured in each case. The application of a CFRP patch on the CSH effectively recovered the strength losses while enhancing the strength in the range of 32% to 45% with respect to the non-strengthened specimens. The developed numerical model based on the cyclic J-integral technique agrees with the test results. This study introduced geometry-related design guidelines for this novel CSH hybrid technique.

**Keywords:** steel members; crack stop hole (CSH); CHS/CFRP hybrid composite; low cycle fatigue; average tensile strength

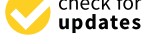



## 1. Introduction

Aged structures are vulnerable to fatigue due to material degradation and heavy loads resulting from increased present service demand. Fatigue usually causes fractures at the microstructural level, which cannot be observed visually [1]. According to the previous investigations, 90% of structural failures occur due to fatigue fractures [2]. However, fatigue is not an engineering problem since it is related to the behaviour of any material due to stress fluctuations. The cyclic effect causes a change in the mechanical properties of a material, which results in failure that occurs below the yield point of the ductile material. Interestingly, quick and sudden failure occurs without any prior warning within a very short duration, which causes the loss of lives and properties due to fatigue. Therefore, fatigue-related repairs to steel structures are very important to continue their services within safety limits. Conventionally, steel plates are connected to cracked members using welding or nuts and bolts as a repair technique. However, these methods show limitations, which include heavy equipment, skilled labour, self-weight increase, change in the microstructure of a material due to heat, discontinuity of the cross section, being difficult to use in an emergency, and high downtime.

Fisher et al., in 1980, proposed a hole placement technique for crack control [3]. The CSH technique has great potential to overcome the majority of the drawbacks mentioned above. This technique could be considered a quick, simple, and cost-effective method. The procedure of the CSH technique is to drill a hole at the end of the crack tip to convert the crack into a notch. The result is a reduction in stress on the cracked structural elements. This technique has been successfully utilized in the aerospace industry since 1950 [4], and, at present, steel bridge repair applications have also introduced this method. However,

the CSH starts re-cracking due to continuous service loads as well as the average strength decline of the cross-section due to the material removal of the CSH. Therefore, there should be a solution to improve the performance of this technique. The size of the CSH is a main debate in regard to the CSH technique and needs an appropriate design guideline to decide the size of the CSH.

A study carried out by Fish et al. recommended a range of the CSH diameter between 50 mm and 100 mm for effective performance [5], while Ishikawa et al. recommended a range of diameter between 6 mm and 25 mm [6]. The minimum hole size should be less than 25 mm for typical highway bridges [3]. Dextor and Oce have confirmed that a minimum size of the CSH is 25 mm in diameter [7]. A formula was developed by Rolfe and Barsom in 1977 for estimating the size of the CSH [8]. Fisher et al. introduced a co-relation to estimate the size of the CSH [3]. This clearly indicates that the size of the CSH is a main debate regarding this technique and that some appropriate design guidelines are needed to decide the size of the CSH under fatigue. Brown et al. suggested that the CSH placed using dull bits would result in fatigue, similar to that of punch holes [9]. CFRP materials have been successfully applied to restore degraded steel structures since 1980 [10] because of their resistance to corrosion, their light weight, and their remarkable fatigue durability [11]. Investigations conducted by Bocciarelli et al. have proven that CFRP exhibits better fatigue performance than welded cover plates methods [10]. Yuana et al. and Liu et al. have also confirmed the fatigue performance of a CFRP-based strengthening technique through their research studies [12,13]. Tang and Kalavagunta et al. have affirmed CFRP's ability to enhance the service life and increase the fatigue capacity of structural members [14,15]. Research studies performed by Kalavagunta et al. have also investigated the steel section's ability to gain strength with CFRP [15]. According to Cadei et al., CFRP sheets and strips could be effectively used in renewing the lost capacity of a material [16]. Gite et al. have shown that CFRP is capable of reducing one third of the weight and strengthening gain by five times with respect to steel [17]. Hence, strengthening the CSH using a CFRP material has great potential to arrest the crack, and it may enhance the fatigue performance due to delays in cracking by nature. This investigation focused on evaluating the fatigue behaviour of a CSH/CFRP hybrid system while introducing a design formula for the CSH technique.

## 2. Methodology

A literature review was conducted to summarize existing investigations related to a crack stop hole as well as CFRP-strengthened techniques and to identify gaps in research. A fatigue loading apparatus was designed and fabricated based on electro-hydraulic controls with a 10 kN load capacity. An experimental test program was performed with four test series to investigate fatigue related measurements of the CSH. The FEM was developed using finite element analysis software (ABAQUS 6.14), with a similar test setup with laboratory tests. The model was used to compare laboratory test results for the purpose of validation. Model results were utilized to conduct a parametric study to estimate the unknown parameter effects on the performance of the CSH. Test results and FEM results were presented with design guidelines and recommendations for the CSH technique.

## 3. Test Setup and Materials

A total of ninety specimens which were categorized into seven series were tested, and the details are listed in Table 1.

The test samples were designed in accordance with ASTM D 790 [18] with the dimensions of 40 mm (width) × 5 mm (thickness) × 280 mm (span) (Figure 1).

**Table 1.** Summary of the test program.

| Series | Geometry | Conditioning Type | No of Samples |
|---|---|---|---|
| $S_1$ | strengthened CSH with varying diameter | Non conditioned | 24 |
| $S_2$ | strengthened CSH with varying diameter | 10,000 cycles | 36 |
| $S_3$ | non-strengthened CSH with change in position | 10,000 cycles | 15 |
| $S_4$ | strengthened CSH with change in position | 10,000 cycles | 15 |

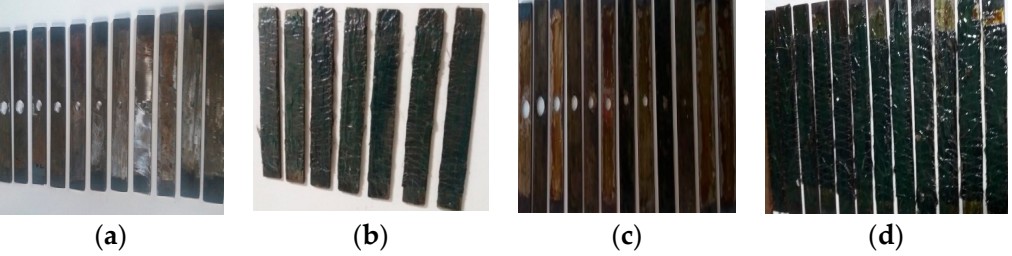

|        (a)        |        (b)        |        (c)        |        (d)        |

**Figure 1.** Prepared specimens of strengthened and non-strengthened CSH: (**a**) non-conditioned, (**b**) non-conditioned, (**c**) conditioned, and (**d**) conditioned.

The surfaces to be strengthened were prepared using sand blasting, and the wet layup method was followed in the strengthening process. The prepared samples were kept curing for 7 days before conditioning (Figure 1). A fatigue load was applied with a 2 kN amplitude and a frequency of 5 Hz within a predefined duration according to ASTM D-790 guidance [18]. After conditioning with a predetermined number of load cycles (Figure 2a), the sample was removed from the fatigue testing machine and fixed to the universal tensile testing machine (UTM) with a 100 kN capacity to apply tensile load, as shown in Figure 2b. This procedure was repeated, and the average value of the tensile load was measured for each specimen. The strain variation at the CSH of the specimen during exposure to fatigue load was recorded using a data logger.

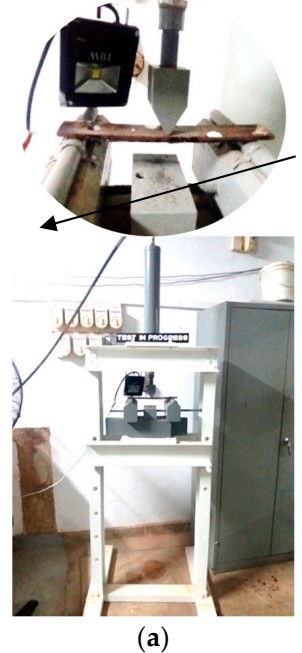
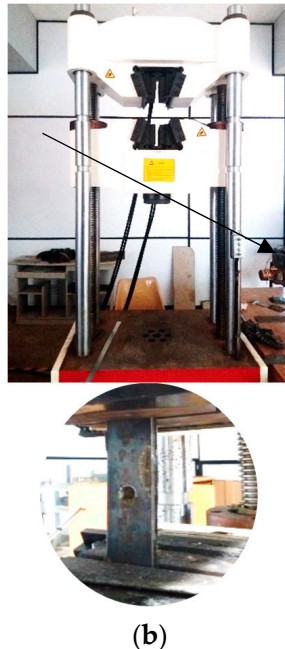

|                    (a)                    |                    (b)                    |

**Figure 2.** Test setup for (**a**) fatigue load and (**b**) tensile load.

The modulus of elasticity and tensile strength of steel and the CFRP material were determined experimentally using a coupon test. The tensile strength and elastic modulus of the steel were measured according to the ASTM D 3039 standards [19]. In fact, a tensile strength of 583 MPa and an average elastic modulus of 200 GPa were reported. Tensile strength and an average elastic modulus of CFRP material were reported at 175 MPa and 1575 MPa, respectively. The manufacturers provided data compared with measured material properties, in accordance with ASTM D 3039/3039 M, which are listed in Table 2.

**Table 2.** Measured and manufacturer-provided material properties [11,20].

| Material Property | Steel | Epoxy Adhesive | CFRP |
| --- | --- | --- | --- |
| Average tensile strength (MPa) | 583 | 25 | 1575 |
| Average elastic modulus (GPa) | 200 | 0.579 | 175 |
| Average Poisson's ratio | 0.3 | 0.3 | 0.3 |

## 4. Results

### 4.1. Effects of CSH Diameter

A total of sixty samples were tested. Twenty-four of them were not exposed to fatigue (non-conditioned) and considered as control samples, and the remaining were conditioned up to 10,000 load cycles with a 2 kN amplitude and 5 Hz frequency. In both strengthened and non-strengthened samples, the ratio between the diameter of the CSH and the width of the member (d/b) varied from 0.1 to 0.6 in a 0.1 step by positioning the CSH at the centre of the midspan. A 200 mm long unidirectional normal modulus CFRP fabric was used for strength. At the end of the fatigue exposure, the test specimen was fixed with a universal tensile apparatus to be measured, the tensile load to be determined, and the average stress to be compared, as shown in Figure 3.

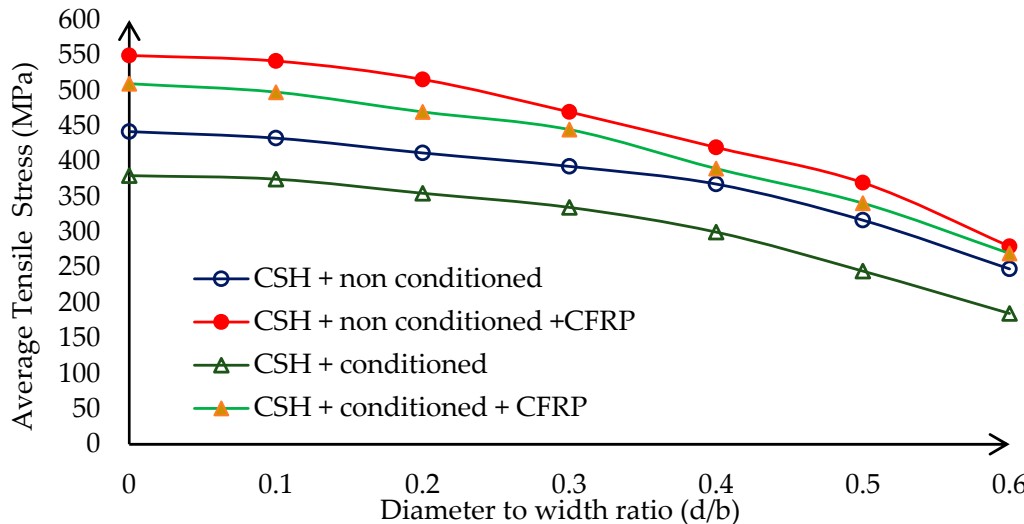

**Figure 3.** Strength losses due to fatigue and strength gains by CFRP.

On average, a 14% strength loss was noted in the specimen due to fatigue effects even without the CSH. This is mainly due to the material degradation caused by repetitive loading and unloading actions (fatigue). On average, 13% to 25% strength loss was noted when the d/b ratio varied from 0.1 to 0.6 at the end of fatigue exposure in the specimens with the CSH. In general, the CSH causes a reduction in stiffness due to the removal of materials. On the other hand, changes in the microstructure of materials near the hole are due to fatigue effects. The change in average tensile capacity of the material is governed by plastic flow effects, imperfections in the crystal lattice of the material, forward and backward motions of dislocations, and plastic deformation along the slip planes of metallic

crystals. In 1954, Coffin and Manson discovered that independently plastic strain was responsible for cyclic damage [21]. All these factors are critically affected by a significant strength loss during fatigue exposure to the CSH. However, the average tensile strength was enhanced by 24% in the CFRP-strengthened CSH system at the end of fatigue exposure. This clearly indicates the ability of CFRP to recover the strength loss with improved fatigue performance. In the suggested system in this study, on average, 32% to 45% strength gain was noted when the d/b ratio varied from 0.1 to 0.6 at the end of the predetermined fatigue exposure. This is mainly due to the composite action and ability of the CFRP laminate to dissipate stress concentrations at the CSH. This reduction can be considered the stiffness lost due to material removal for the CSH, while the strength losses are due to fatigue. Therefore, it has the potential to enhance stiffness as well as restore the fatigue-bearing capacity of the material at the same time. Therefore, the proposed CSH/CFRP hybrid repair technique has the potential to enhance the stiffness of the member as well as the fatigue-bearing capacity of the material. Failure modes observed during the tensile testing of these series are shown in Figure 4. The variation in strain near the crack stop hole was monitored during fatigue load application. Conventional strain gauges are attached to the steel surface, 5 mm away from the periphery of the CSH. The strain near the CSH at the end of each pre-determined load cycle is plotted in Figure 5.

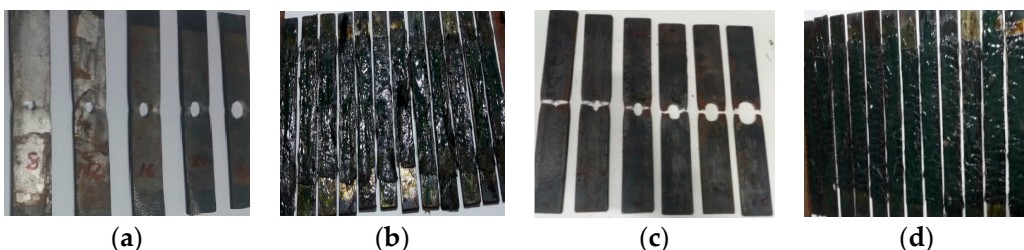

**Figure 4.** Failure mode of the CSH; (**a**) non fatigue and non-strengthened, (**b**) non fatigue and strengthened (**c**) fatigue and non-strengthened (**d**) fatigue and strengthened.

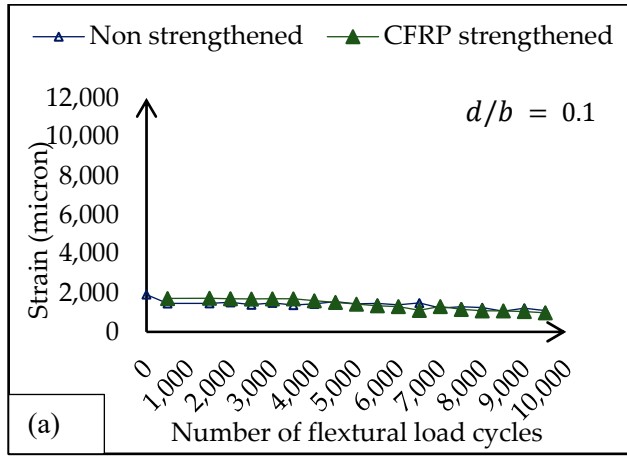
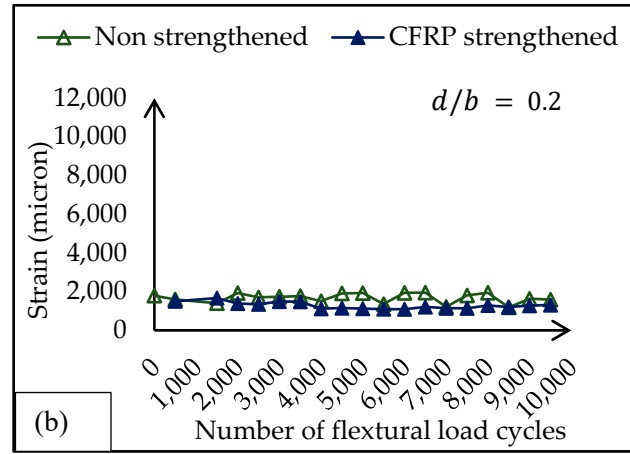

**Figure 5.** *Cont.*

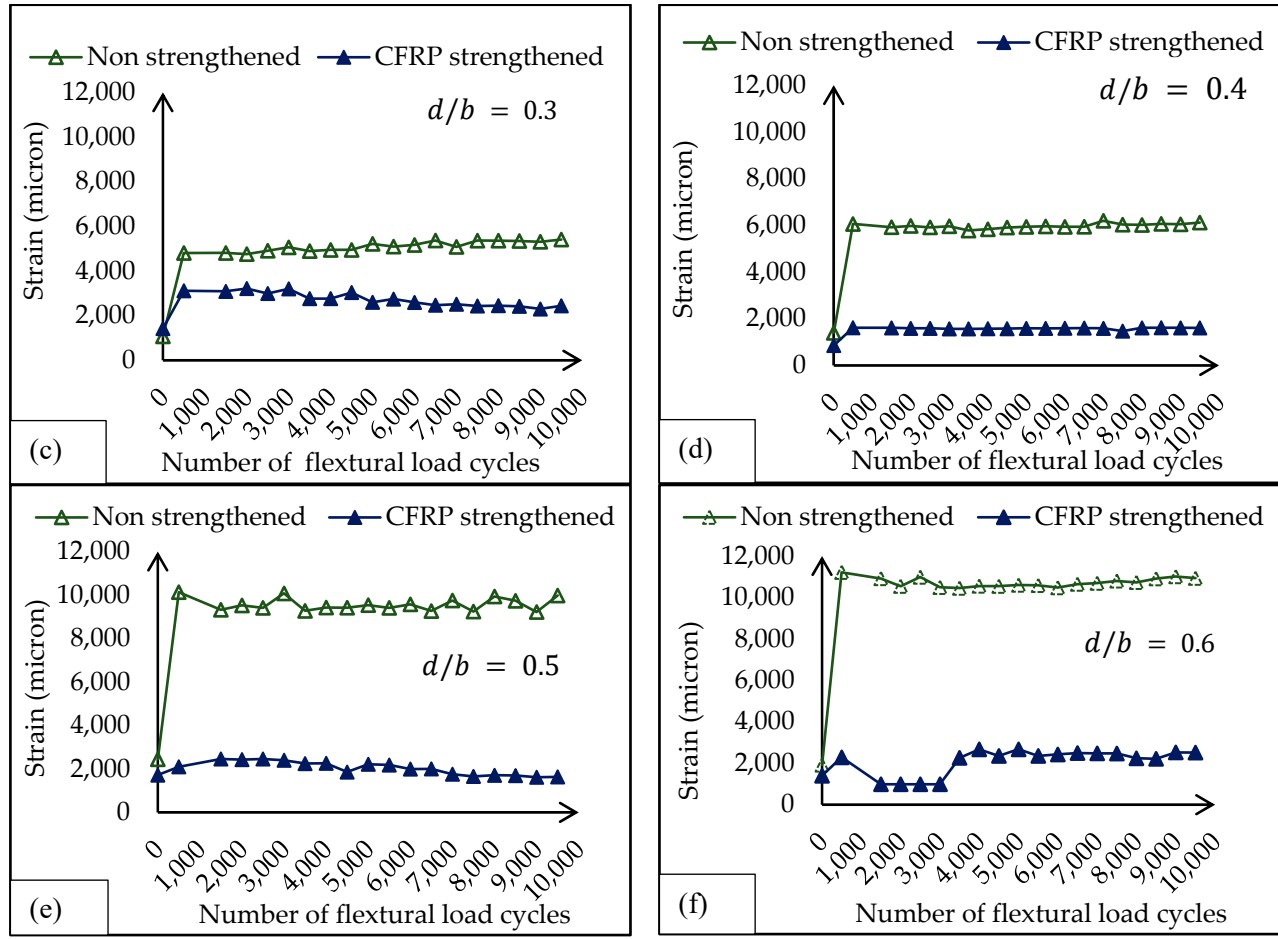

**Figure 5.** Strain variations during conditioning when $d/b$ is (**a**) 0.1, (**b**) 0.2, (**c**) 0.3, (**d**) 0.4, (**e**) 0.5, and (**f**) 0.6.

According to Figure 5a,b, no significant deviation was noted in strengthened and non-strengthened samples. The contribution of the CFRP material is negligible, and the steel substrate can withstand the effects of fatigue. When the d/b ratio increases from 0.3 to 0.6, the strain variation on the steel substrate of samples with CFRP indicates a significant strain reduction compared to the non-strengthened CSH, as shown in Figure 5c–f. This is due to the effective load-sharing ability of the attached CFRP layer. The results in strain control contribution, further delaying crack propagation in the CSH. Hence, measured strain variation indicates the effectiveness of the proposed hybrid repair technique for steel members exposed to fatigue over the traditional CSH technique.

### 4.2. Effects of the Offset Distance of CSH

A total of 36 steel specimens with a change in position of the CSH were conditioned up to 10,000 load cycles with a 2 kN amplitude and 5 Hz frequency. Eighteen of them were reinforced with a 200 mm long CFRP layer. The remaining samples were non-strengthened and taken into consideration as controlled specimens. In these two test series, the distance from the loading point to the CSH was considered the main variable, and a CSH of a 16 mm diameter was placed at the pre-determined locations of the member. The distance to the CSH from the midspan varies from 20 mm to 100 mm in 20 mm steps. At the end of fatigue loading, the average tensile strength of the samples was determined using a universal tensile testing apparatus, and the results are plotted in Figure 6.

The main purpose of this test series was to compare the retained tensile strength after fatigue load exposure to quantify the effectiveness of CFRP on the distance to the CSH from the mid-span. A trend of average retained tensile strength variation deviation with

respect to the ratio between the diameter and offset distance of CSH was noted from both strengthened and non-strengthened samples. When the offset distance increases the stress concentrations near the CSH are reduced. The graphs also provide convincing evidence of such behaviour. The CFRP layer helps to reduce the stress concentration while enhancing the tensile strength of the specimen. As a result, the CSH/CFRP hybrid system showed a significant strength gain compared to similar non-strengthened specimens, which restored the strength loss due to the opening of the CSH. The fatigue-sensitive zone is near the loading point, and this region can be considered a high-stress area of the specimen because the flexural cyclic load helps to increase the stress intensity near the midpoint. This clearly indicates the extension of the fatigue performance of steel members when applying the suggested hybrid repair technique in this study, irrespective of the location of the CSH, with guaranteed performance compared to the conventional CSH technique. Failure mode was observed as the de-lamination of the CFRP layer.

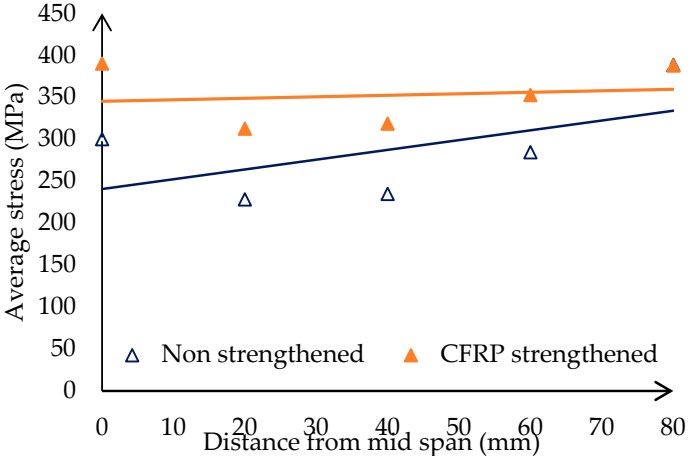

**Figure 6.** Comparison of average strength variation with CSH position.

Strain gauges were fixed near the specimens, as shown in Figure 7. The average strain variation during fatigue was measured at the end of each pre-determined load cycle, as shown in Figure 8, based on the distance to the CSH from the mid-point. When the CSH is near the midspan, it is subjected to heavy stresses, which dissipate through the CFRP sheet. As a result, there is a comparatively lower strain in the steel substrate of the hybrid arrangement when compared with the conventional CSH arrangement. When the position of the CSH moves from the centre to the support, the fatigue stress also reduces. Hence, similar strain variations were observed in both strengthened and non-strengthened samples. This clearly indicates that the proposed CSH/CFRP hybrid arrangement could successfully accommodate the strain near the CSH.

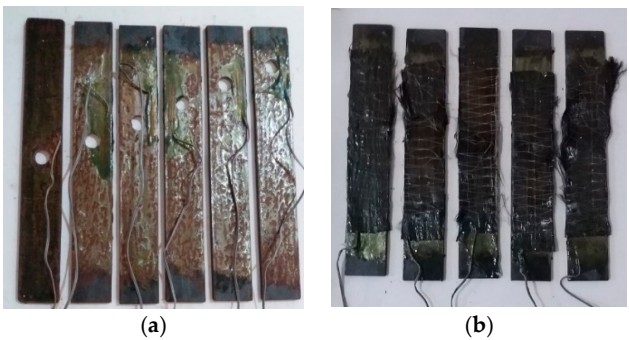

**Figure 7.** Strain gauges attached, (**a**) non-strengthened and (**b**) CFRP-strengthened specimen with offset CSH.

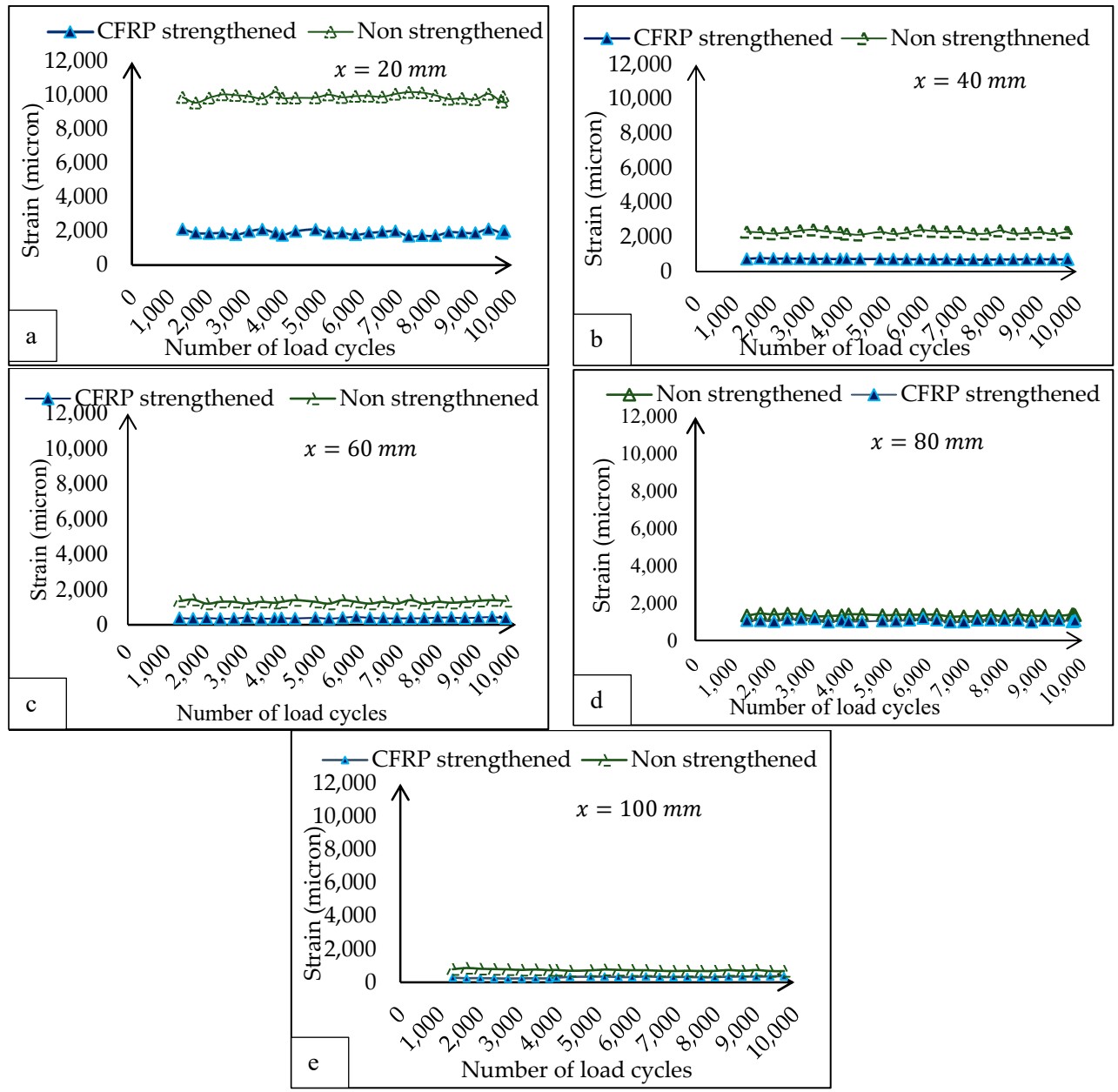

**Figure 8.** Strain variations with offset distance of CSH when *x* is (**a**) 20 mm, (**b**) 40 mm, (**c**) 60 mm, (**d**) 80 mm, and (**e**) 100 mm.

## 5. Finite Element Modelling (FEM)

Elastic plastic fracture mechanics (EPFM) could be considered as an alternative fracture mechanics model. This theory is used to analyse fractures caused by large deformations at the crack initiation stage. When the large plastic zone is formed in front of the crack end, the theory of EPFM can be applied, which explains the crack tip plasticity. In addition, elastic plastic fracture mechanics implies time-independent materials, and the plasticity effect on the crack tip is taken into account in this approach. A numerical model was developed to simulate the performance of the CSH/CFRP hybrid technique using of commercially available finite element software (ABAQUS 6.14). The geometrical configuration, boundary conditions, material properties, and attributes were the same as in the test program. A general contact standard option of the ABAQUS was utilized with mechanical friction to simulate the interfaces between two supportive rollers and the bottom surface of the plate and the loading nose on the mid-plane of the steel plate. Each specimen model was loaded

with the LCF mode using a direct cyclic option and an input of 10,000 cycles. When the size of the mesh is very small, it results in the generation of a singularity point. The stress at the singularity points is theoretically considered to be infinity. Therefore, considering all of the above factors, a FEM result of 2 mm mesh size was selected as an appropriate size in this analysis, as shown in Figure 9.

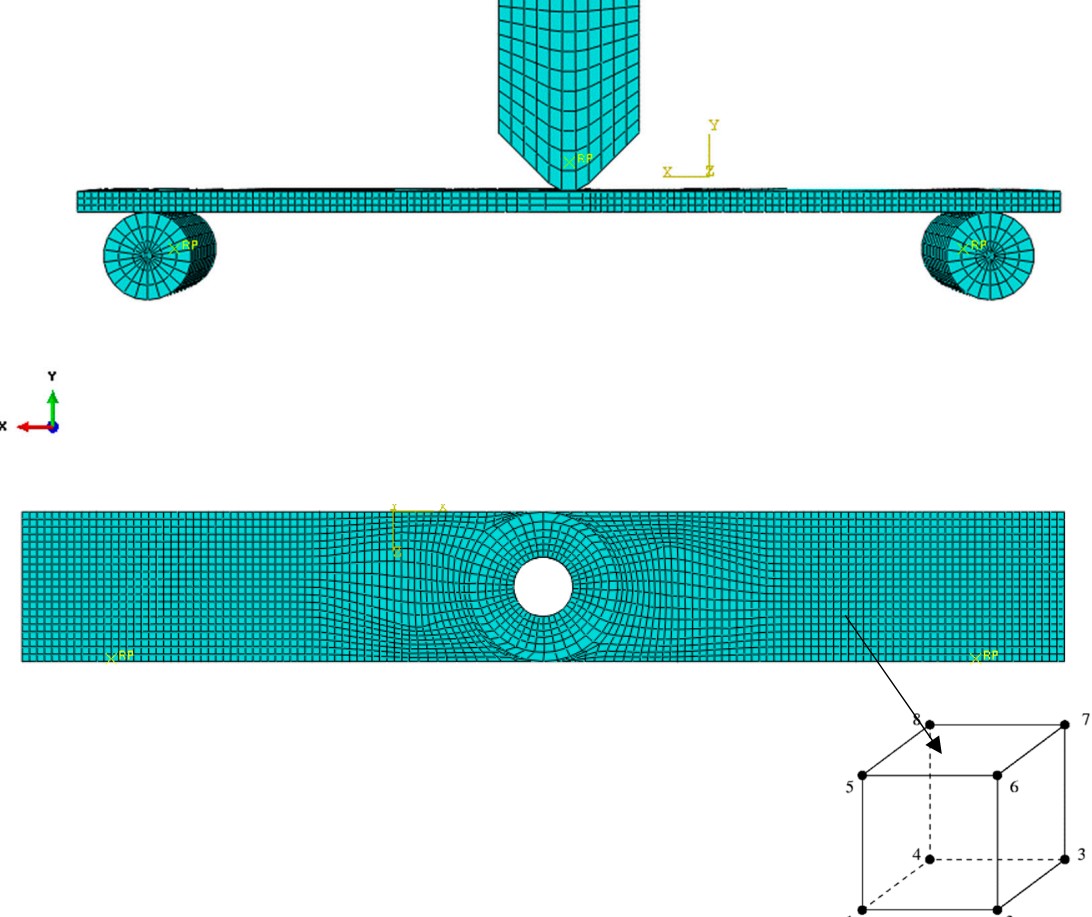

**Figure 9.** The assembly model of the specimen.

The magnitude of pressure applied on top of the loading nose was selected as 1.25 N/mm$^2$ (2 kN), and the frequency of the loads was fixed at 5 Hz. When considering the boundary conditions, two supportive cylinders were fixed to avoid both rotation and translation. The loading rate was kept constant at a fixed rate of 0.2 mm/min throughout the analysis. The Hashin damage option was used to model the CFRP fabric. The coefficient of the friction value was taken as 0.15 between each surface. Special options were used to create the crack, while the extension direction was selected using the q vector option. The model had the same configurations as the laboratory test setup. Each specimen module was loaded with the LCF mode using a direct cyclic option. An 8 mm radius loading nose was used to apply loads at the mid-plane of the specimen. The steel plate with the CSH was modelled using elastic–plastic option with isotropic hardening condition. Three-dimensional eight-node solid brick elements (C3D8R) were used to model the steel plate. All the material properties assigned to the model were the same as in the test procedure explained in Table 2. Therefore, computing the time and improving the level of accuracy with mesh size should be optimized in this situation. Mesh refinement is the most vital method of judging the mesh size, and it can be classified as coarse mesh, medium mesh, or fine mesh. Reducing the element size is the easiest mesh refinement strategy, with element sizes reduced throughout the modelling domains. This approach is attractive due

to its simplicity, but the drawback is that there is no preferential mesh refinement in regions where a locally finer mesh may be needed. Therefore, the need to select a fairly good mesh size for the proposed FEM with the required level of accuracy becomes important. In industrial practice, a fine mesh is introduced only around sensitive regions of high stress concentrations, and the remaining area, which has a lesser importance, is introduced to a coarser mesh. The convergence test is another way to check whether the size of the mesh is worth it or not for the FEM. In this method, half of the mesh size is selected and compared with the previous analysis, and the program is re-run with the new analysis. If the results are insignificant when compared to the previous mesh size, it is considered an appropriate mesh size. This could be repeated until the level of acceptance. However, there are no definitions regarding the mesh size when starting this convergence test, and it should be repeated at least three times with different mesh sizes until there are insignificant variations. High stress near the CSH area displays a higher level of convergence than other areas. During analysis, attention should be on the stress singularity because it does not allow it to converge. When the size of the mesh is very small, it results in the generation of a singularity point. Stress at the singularity points is theoretically considered to be infinity. Therefore, considering all of the above factors, an FEM result of 2 mm mesh size was selected as an appropriate size in this analysis. It was confirmed by the mesh sensitivity analysis and the results of the mesh sensitivity analysis. The composite lay-up option of the material property model was utilized to embed the CFRP layer in this regard. The conventional shell elements and material type were selected as the lamina type. The Hashin damage option was used to model the CFRP fabric, and a tie constraint was used to represent the bond between steel and the CFRP. The assembled mesh model was run for analysis, and the results were collected from the ODB file in the visualization module at the end of the analysis. FEM results were compared to validate the developed model. The contours of stress variation with diameter for non-strengthened and CSH/CFRP hybrid systems are shown in Figure 10a,b, respectively.

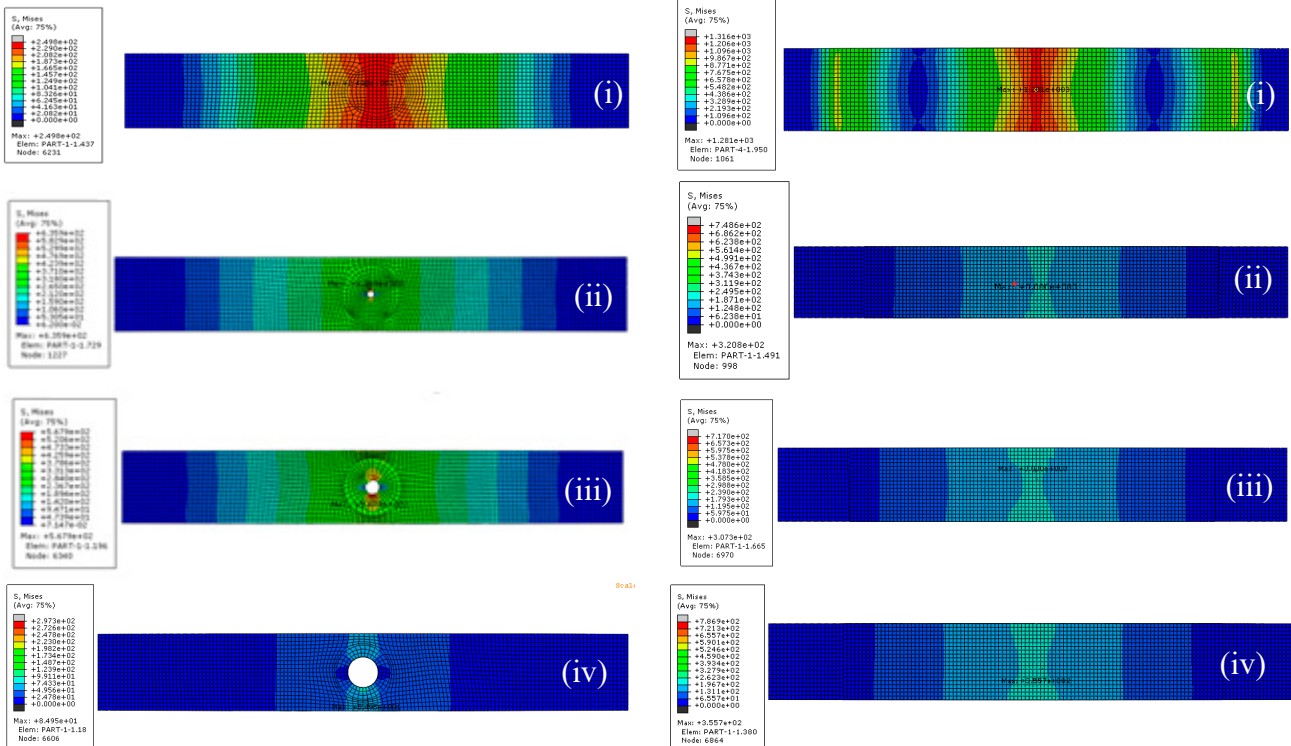

**Figure 10.** *Cont.*

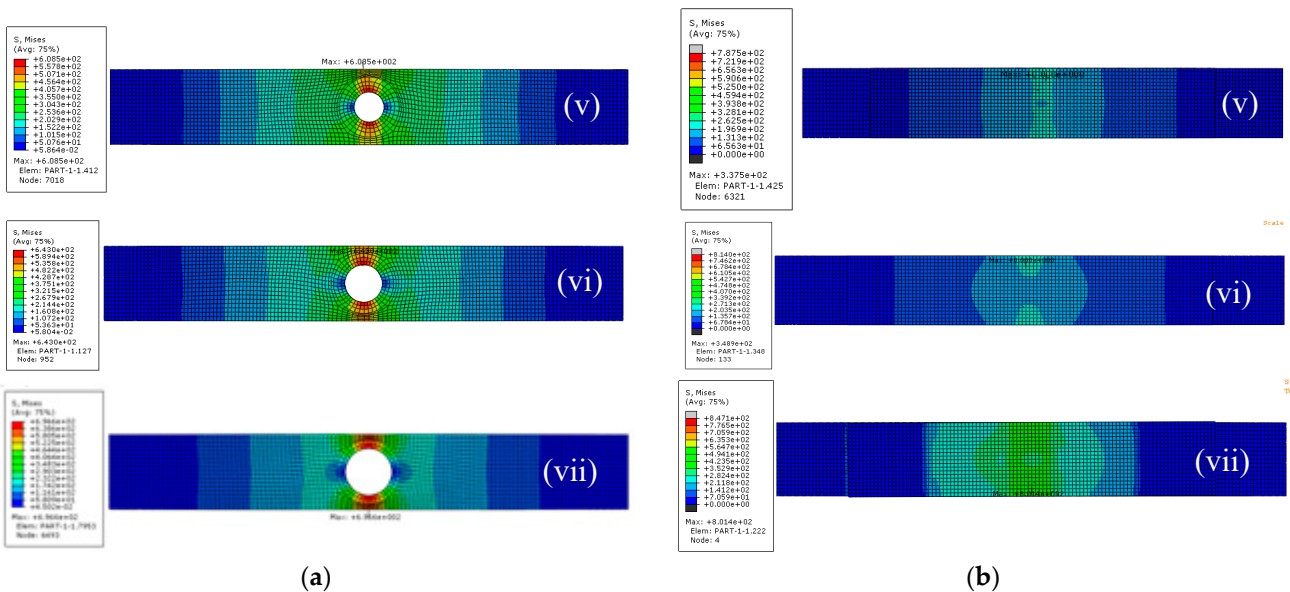

**Figure 10.** Contours of stress variation with diameter-to-width ratio for (**a**) non-strengthened, (**b**) CSH/CFRP hybrid: (**i**) 0, (**ii**) 0.1, (**iii**) 0.2, (**iv**) 0.3, (**v**) 0.4, (**vi**) 0.5, and (**vii**) 0.6.

The effect of the diameter-to-width ratio of the CSH was taken into consideration as a primary variable in this analysis. The predicted results for non-strengthened and CSH/CFRP hybrids (after fatigue exposure) were compared with relevant laboratory test results, as shown in Figure 11. Contours of stress variation with offset distance for non-strengthened and CSH/CFRP hybrid systems are shown in Figure 12a,b, respectively. According to the graphs, the model results and experimental results were reasonable for the selected range of the diameter-to-width ratio of the CSH. On the other hand, the strain variation due to the diameter-to-width ratio of CFRP-strengthened CSH test results agreed well with the FEM results, as shown in Figure 11.

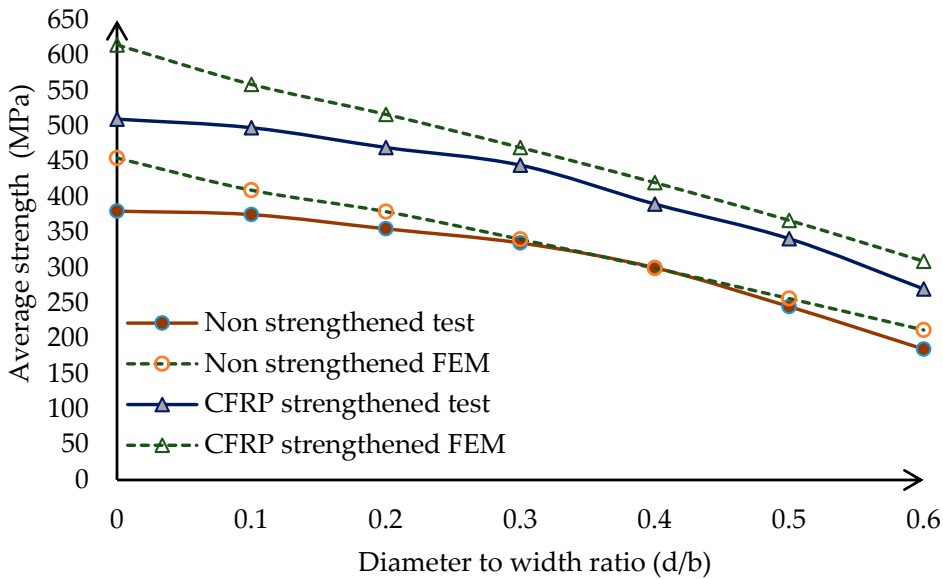

**Figure 11.** Comparison between experimental and FEM results.

The effect of the offset distance of the CSH was taken into consideration as a primary variable, and it varied from 0 to 80 in 20 mm steps. The predicted results belonging to non-strengthened and CFRP-strengthened groups (after fatigue exposure) were compared with relevant experimental results, as shown in Figure 13.

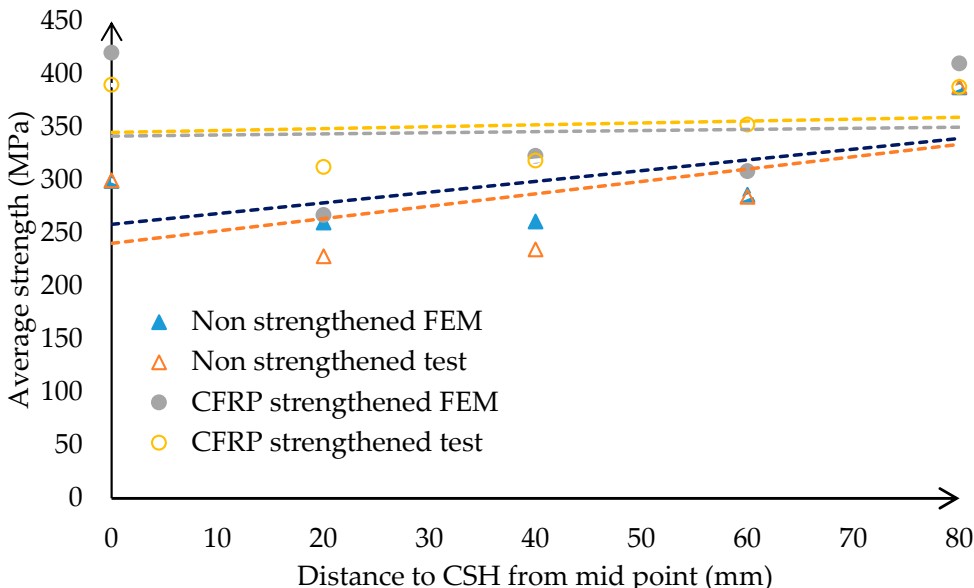

**Figure 12.** Contours of stress variation with offset distance for (**a**) non-strengthened, (**b**) CSH/CFRP hybrid system: (**i**) zero, (**ii**) 20 mm, (**iii**) 40 mm, (**iv**) 60 mm, and (**v**) 80 mm.

**Figure 13.** Comparison between experimental and FEM results.

The model-predicted results of the conditioned CSH samples indicate, on average, a 5% and 7% difference to tested results for non-strengthened and CFRP-strengthened samples, respectively, in Figure 13. When the distance to the CSH from the mid-point varies, the average deviations between the predicted and tested results of non-strengthened and CFRP-strengthened samples are 7% and 8%, respectively. This clearly indicates the accuracy of the developed models that could be used to further simulate the behaviour of the suggested CSH/CFRP hybrid system.

### 5.1. Validation

The mechanical properties of the material play a vital role when considering fatigue. Yield strength and tensile strength are the two main material properties considered in the fatigue analysis of ductile material. These material properties represent resistivity to failure due to deformation and fracture, respectively. Yield strength and tensile strength represent different meanings related to fatigue. Yield strength represents maximum stress just before permanent deformation, and tensile data represent maximum stress prior to material fracture. When considering the ductile material yield value, it is prominent, and the brittle material deals with tensile strength. As brittle material does not achieve the yield load, when performing tensile loads, it suddenly recedes to fractures. Hence, this study used a tensile test to determine the yield strength (fatigue stress) of the CSH at the end of the condition to measure any retained yield strength.

This study retained an average yield stress of the material (ratio of yield load to cross-section area), which was estimated at the end of the fatigue loads as retained yield strength, which indicates the effects of fatigue on the CSH specimen. The developed FEM also estimated yield strength at the end of fatigue, and it was measured using stress at the edge of the CSH. This value is nearly three times the applied load on the specimen. This approximation is an extension of the theory developed by Inglish in 1913. The author has introduced a stress calculation technique closer to the hole. In addition, the stress–strain curve under the visualization model in the field output data could also be utilized to estimate the yield stress. The strain distribution near the CSH was measured and compared with FEM results, as shown in Figures 11 and 13. According to the graphs, the model results and experimental results were reasonable for the selected range of the CSH. The strain variation in CFRP-strengthened CSH test results agreed well with the FEM results.

### 5.2. Parametric Study

A numerical model was utilized to investigate the parametric influences on the CSH. In this study, member thickness, crack length, and offset distance were selected as critically influencing the geometrical parameters in the range considered. According to the experimental program, the diameter of the CSH and the position of the CSH effects were experimentally investigated, and test results have confirmed significant effects for those parameters. The results from this study could be summarized as follows:

- This investigation reported the strength losses which were in the range of 13% to 25% compared to the non-conditioned CSH with the diameter ranging from 4 mm to 25 mm.
- This investigation reported the tensile strength enhancement with CFRP which is in the range of 32% to 45% compared to the non-strengthened CSH with the diameter range change from 4 mm to 25 mm.
- This study recorded a strength increase with respect to off-set distance, which was in the range of 36% to 131% compared to the CSH at the midpoint.
- CFRP-strengthened CSH reported significant strength gain and it's variation were recorded as 19% to 42% with respect to offset distance.
- Failure mode was observed as de-lamination of the CFRP layer with the fatigue load as well as the tensile load.

- A sound numerical model was developed based on material properties capable of simulating the fatigue behaviour of the CSH subjected to cyclic flexural loading.
- Design guidelines for the CSH technique summarized in Table 3.

**Table 3.** Fatigue-related design guidelines for CSH.

| Category | Parameter | Range of Operation |
|---|---|---|
| Geometry | Diameter to width ratio (d/b) | $\frac{d}{b} \leq 0.2$ |
| | Thickness to diameter ratio (t/b) | $\frac{t}{d} \leq 0.5$ |
| | Diameter to crack length ratio (d/L) | $\frac{d}{L} \geq 0.4$ |
| | Diameter to offset distance ratio (d/x) | $\frac{d}{x} \leq 0.2$ |

## 6. Engineering Implications

According to results from the laboratory test, the FEM results used to implement design guidelines for the CSH technique. This method could replace conventional fatigue repair techniques and re-install the strength losses of aging structures. Ultimately, it may contribute to delays in repairs or replacements as well as inspection intervals. In addition, this may overcome drawbacks related to conventional repair methods. As a result, it contributes to saving money with continuous service demands according to the present demand.

## 7. Conclusions and Recommendations

The detailed experimental and numerical simulations conducted on the developed novel hybrid technique to control cracks in steel elements in civil engineering infrastructures yielded the following conclusions:

- Introduced design guidelines to decide the appropriate size of the CSH for crack control on steel structures due to fatigue.
- The CSH effectively recovers the strength losses in the range of 32% to 45% with respect to the non-strengthened condition.
- The proposed method exhibits a significant capacity enhancement for crack controls, and it is strongly recommended to improve the service life of steel infrastructures under fatigue load.
- This developed numerical model can be effectively utilized to estimate the effects of unknown parameters of the CSH under a fatigue response.

## 8. Future Works

Fatigue is governed by the microstructure of a material due to external or internal stress. Therefore, there should be a microscopic analysis regarding grain arrangement, dislocation, and information regarding the missing atoms of the material with critical parameters. The finite element model does not have the facility to model the effects of environmental factors on bond performance. This could be considered a significant gap in FEM techniques. Furthermore, evaluating the long-term bond performance-related investigations of the CSH/CFRP hybrid system to confirm the durability of this technique could also be carried out in the future.

**Author Contributions:** Conceptualization, S.A. and J.C.P.G.; methodology, S.A.; software, S.A.; validation, S.A., J.C.P.G. and S.F.; formal analysis, S.A.; investigation, S.A.; resources, J.C.P.G.; data curation, S.A.; writing—original draft preparation, S.A.; writing—review and editing, J.C.P.G.; visualization, S.F.; supervision, J.C.P.G.; project administration, J.C.P.G. All authors have read and agreed to the published version of the manuscript.

**Funding:** This research received no external funding.

**Data Availability Statement:** Data are contained within the article.

**Acknowledgments:** We would also like to express my gratitude to all academic staff members in the Department of Civil Engineering, University of Moratuwa, Sri Lanka, for their valuable support. D.M.N.L. Dissanayaka, technical officer in the Structural Testing Laboratory, Yohan technical officer in the Computational Mechanics Laboratory, Charaka Satharasinghe, technical officer in the Computer Laboratory for their valuable support extended throughout the research period. Moreover, We would like to thank Airow Solution (PVT) Ltd. for material providing and Randana de Silva, Pramod Karunarathna, and Varakini for their kind support throughout this period. Finally, We wish to acknowledge Thanuja and Sanjalee Abeygunasekara, for their love, emotional support and encouragement. The authors gratefully acknowledge the support of the staff in the structural testing laboratory, as well as the staff in the computer laboratory, Department of Civil Engineering, University of Moratuwa, Sri Lanka, for their valuable support.

**Conflicts of Interest:** The authors declare no conflict of interest.

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
