# Peer review of "Investigation of Crack Repairing Technique to Delay Fracture Initiation of Steel Members Subjected to Low Cycle Fatigue"

_buildings, doi:10.3390/buildings13122958_

Round 1

Reviewer 1 Report

Comments and Suggestions for Authors

The authors have presented a study about investigating on crack repairing technique to delay fracture initiation of steel members subjected to low cycle fatigue. They have used finite element analysis using ABAQUS software, however, the following points need to be explained:

1) the material properties are not provided.

2) defining plasticity in ABAQUS

3) elements type

4) specifying tangential behavior 

The authors have not shown the way they have modeled fatigue loading. 

CFRP properties are not provided.   

When the short cracks growth, the fatigue notch sensitivity must be an issue that needs to be addressed. The author(s) have not shown any mention to it. 

How did the authors simulated  the number of load cycles in ABAQUS?

The authors have conducted a total of hundred and sixty-two specimens for their experimental tests, whereas ONLY two tests were considered for their FEA study, WHY? 

Comments on the Quality of English Language

Line (11) enhance, must be " enhances".

Lines (28,29 and 30) have gaps between the words!

Author Response

Dear Sir

I addressed all of your valuable comments on manuscript 2643543 (building)

Thanking You

Dr.Sampath 

Reviewer 2 Report

Comments and Suggestions for Authors

Although, the submitted research work on “Investigation on crack repairing technique to delay fracture initiation of steel members subjected to low cycle fatigue” appears to be an interesting work. However, there are certain issues with the same. Author(s) could be provided an opportunity to resubmit. Following are initial comments:

1. The discussed literature studies in the introduction section are less in order to establish a research problem. Suggest adding a few more and explaining them in brief too. Therefore, problem formulation from the state of art literature as presented in introduction section is not satisfactory. It needs to be explained properly in an expanded form.

2. Table 1 in present form is confusing.

3. There is no need of showing basic experimental set up of fatigue and tensile. If they are to be kept, proper labelling and quality of images should be taken care of.

4. Related to comment 2, Table 2 also has been presented in a vague form. Which value in a specific row corresponds to is completely confusing.

5. It would have been better if experimental illustrations also included digital image correlation (DIC) results or subsequent fractographic results. Do author(s) have these results? If yes, they should be added to enhance analysis and establish strong scientific evidence on fracture initiation of steel members subjected to low cycle fatigue.         

6. Quality of Figure 15 is unacceptable. Assembly model should clearly discern all parts.     

7. FEM section 4 is not discussed well. It should clearly bring out or indicate validation points.   

*Author(s) should highlight all the modifications carried out in the paper.

Author Response

Dear Sir

I have addressed all of your valuable comments of the manuscript 2643543 

Thanking You

Dr.Sampath

Reviewer 3 Report

Comments and Suggestions for Authors

The conducted work “Investigation on crack repairing technique to delay fracture initiation of steel members subjected to low cycle fatigue” is good. However, following comments should be addressed to further improve the paper:

A. GENERAL COMMENTS TO IMPROVE PAPER

1.     Explicitly mention the novelty and research significance of current work in last paragraph of introduction section with emphasis on scientific soundness. Also, it would be great to add recent relevant literature review from more 2023 papers in introduction section as there is no paper cited from 2023.

2.     Avoid paragraph of few (i.e. 1-4) sentences throughout the manuscript, particularly in results section, e.g. lines 27-32, 192-193, 198-199, 341-342, etc.

3.     Texts within figures must be clearly readable.

4.     There are too many figures. Only very important figures may be retained in main text body. And important figure may be shown in annexure.

5.     Results are very briefly explained/elaborated in a descriptive way, thus results in current form look like a project report. Results should be further elaborated with scientific reasoning.

6.     A separate brief section (explaining the relevance of this research for practical implementation) may be added before conclusion section.

7.     Conclusions are little long; these should be to the point with some quantitative way as obtained from results with scientific soundness. Closing remarks should be added at the end of conclusion section keeping in mind all conclusive bullet points.

8.     English Language should be improved throughout the manuscript.

B. SPECIFIC COMMENTS TO IMPROVE FOCUSSED RESEARCH

1.     Please provide rationale of crack repairing technique.

2.     Please label figures 2, 4, 6, 7, 10, 11 and 12 in such a way to tell reader how to conceive it.

3.     Please provide tabulated data in support of figure 15, i.e. which input parameters are being used.

4.     Lines 403-405: how is the developed model validated for predicting precise behavior?

5.     Line 387: how much is the difference between experimentally and numerically estimations?

Comments on the Quality of English Language

Avoid paragraph of few (i.e. 1-4) sentences throughout the manuscript, particularly in results section, e.g. lines 27-32, 192-193, 198-199, 341-342, etc.

English Language should be improved throughout the manuscript.

Author Response

Dear Sir

I have addressed all of your valuable comments on manuscript 2643543 (building)

Thanking You

Dr.Sampath  

Round 2

Reviewer 1 Report

Comments and Suggestions for Authors

Aceppted! 

Reviewer 2 Report

Comments and Suggestions for Authors

The revised version is satisfactory. 

Author Response

Dear Sir,

Thanks for your valuable comments and contributions of regarding this paper